# Research Progress of Rapid Non-Destructive Detection Technology in the Field of Apple Mold Heart Disease

**DOI:** 10.3390/molecules28247966

**Published:** 2023-12-06

**Authors:** Yanlei Li, Zihao Yang, Wenxiu Wang, Xiangwu Wang, Chunzhi Zhang, Jun Dong, Mengyu Bai, Teng Hui

**Affiliations:** 1Mechanical and Electrical Engineering College, Beijing Polytechnic University, Beijing 100042, China; yzh05201900729@163.com (Z.Y.); wxw020619@163.com (X.W.); zcz@bgy.edu.cn (C.Z.); 1703@bgy.edu.cn (J.D.); 18032992591@163.com (M.B.); 2Food Science and Technology College, Hebei Agricultural University, Baoding 071001, China; cauwwx@hebau.edu.cn; 3Food Science College, Sichuan Agricultural University, Ya’an 625014, China; huiteng@sicau.edu.cn

**Keywords:** apple, mold heart disease, non-destructive, spectroscopy, intelligent

## Abstract

Apples are rich in vitamins and dietary fiber and are one of the essential fruits in people’s daily diet. China has always been a big apple consumer, and with the improvement of people’s life quality, nutrition, and health requirements, the demand for high-quality apples has increased year by year. Apple mold heart disease is one of the main diseases affecting apple quality. However, this disease cannot be easily detected from the surface, so it is difficult to detect mold heart disease. Therefore, this paper focuses on the analysis of seven non-destructive detection technologies, including near infrared spectroscopy technology, hyperspectral technology, Raman spectroscopy technology, electronic nose technology, acoustic technology, electrical technology, and magnetic technology, summarizes their application status in the detection of apple mold heart disease, and then analyzes their advantages and disadvantages. Combined with the current rapid development of artificial intelligence (AI) technology, this paper proposes the future development trends of using non-destructive technologies to detect apple mold heart disease. It is expected to provide basic theory and application references for the intelligent detection of apple mold heart disease.

## 1. Introduction

Apple has a high nutritional value and is rich in dietary fiber, vitamins, and a series of nutrient compositions [1,2]. Apple is a fruit produced in many countries. With the continuous improvement in humans’ life quality and the increase in daily consumption of fruits, the demand for high-quality apples is growing rapidly [3].

Apple mold heart disease, seriously affecting apple quality and yield, is a major disease that is caused by a variety of bacteria, such as Penicillin and Patulin. There are five to six types of fungi that have the highest frequency of occurrence among the apple core-dwelling bacteria, including Alternaria, Pink Monotelomeria, Fusarium, and Pseudomonas. The apple suffers from damage starting from the ventricles and gradually spreading outward to rot. The heart of the diseased apple turns brown and is filled with gray-green mold-like substances (mainly composed of Alternaria and Fusarium), sometimes pink mold-like substances (mainly composed of Pink Monocytospora), and these pathogens all have the ability to produce toxins. During storage, when the core of the apple is severely moldy and rotting, water-stained, brown, and irregularly shaped wet rot plaques can be seen on the apple carcass [4,5]. The plaques can be connected to each other and form patches [4,5]. Finally, the entire apple rots, and the flesh has a bitter taste. Especially when diseased fruits are mixed into the deep processing of apples, the moldy tissue that accumulates toxins can lead to excessive accumulation of mycotoxins in deep processed foods such as apple juice, jam, and canned food (the World Health Organization sets a maximum concentration of 50 μg/L), causing potential food safety hazards [6,7,8]. If people consume the fruits of this disease, the risk of cancer, teratogenicity, and fertility will greatly increase, posing a serious threat to human health [9,10]. The symptoms of diseased apples mainly include fruit ventricular mildew (mildew heart type), fruit ventricular browning (browning type), and fruit heart rot (heart rot type) [11,12]. The diseased apple entering the market will seriously endanger the health of consumers and affect the economic benefits of the apple enterprise. The mildest symptom of mold heart disease is that it only occurs within the fruit pit and does not infect the flesh, and the pit area generally accounts for 6% of the cross-sectional area of the apple. Therefore, a disease degree of less than 6% is considered the standard for mild mold heart disease. In some relevant literature, a degree of less than 10% is defined as mild mold heart disease [13], and a degree of 6% to 10% is defined as slightly severe mild mold heart disease. In addition, the incidence rate of 10% to 14% is general mold heart disease, and when the incidence rate is greater than 14% of the affected area, the apple has already undergone severe mold transformation, which is easy to distinguish from the surface of the apple. As the severity of the disease increases, the accuracy of identifying mold heart disease shows an upward trend. Samples of apples with different degrees of mold heart disease are shown in Figure 1. However, the surface of the diseased apple shows no obvious external symptoms in the early and middle stages of mold heart disease, which causes difficulties in picking a good apple. Therefore, a non-destructive detection method for apple mold heart disease is extremely important for the development of the world’s apple industry.

At present, the methods for detecting apple mold heart disease mainly include traditional detection methods and rapid, non-destructive detection methods. The traditional detection methods include subjective detection and biochemical detection, both of which have the disadvantage of being tedious, long, and expensive. Rapid non-destructive detection methods mainly include spectroscopy technologies, imaging technology, low-frequency magnetic resonance, and so on, which are highly favored by many scientific personnel because of their advantages of detection rate, no sample destruction, and non-pollution. In recent years, rapid non-destructive detection technologies have been developing rapidly, showing significant development prospects in numerous fields [14]. In order to explore the feasibility of these technologies in detecting apple mold heart disease, many researchers have used and compared various rapid, non-destructive detection technologies [15].

This paper reviewed the research progress of rapid non-destructive detection technologies for apple mold heart disease in recent years, briefly introduced the principles of these technologies, summarized the advantages and disadvantages of detection of apple mold heart disease, and finally proposed the challenges faced and development prospects in practical application combined with the current rapid development of artificial intelligence (AI) technology. It is expected to provide a basic theory and application reference for apple mold heart disease intelligent detection for the global apple industry.

## 2. Application Progress of Rapid Non-Destructive Detection Technologies for Apple Mold Heart Disease

The rapid non-destructive detection technologies mainly include near infrared (NIR) spectroscopy technology, hyperspectral imaging (HSI) technology, Raman spectroscopy technology, electronic nose technology, acoustic technology, electrical technology, and magnetic technology [16]. In recent years, rapid non-destructive detection technologies have increasingly been applied in fields of fruit and vegetable quality and safety [17,18,19]. Numerous researchers have explored the feasibility of these technologies for apple mold heart disease [20,21,22].

### 2.1. Spectroscopy Technology

Thanks to the rapid development of computer technology, big data technology, and optics technology, spectroscopy detection technologies have been widely used in detecting the quality of apples [23,24,25,26]. The basic principle of spectroscopy detection, an analytical method, is to determine the chemical composition and substance structure by determining the characteristic wavelength and intensity of light absorption, emission, or scattering. The interaction of light with matter changes with wavelength, and different substances absorb, emit, or scatter light in different wavelength regions [15].

#### 2.1.1. NIR Spectroscopy

NIR spectroscopy is the most commonly used spectroscopic technology. NIR is an electromagnetic radiation wave (780–2526 nm region) between the visible spectrum (VIS) and mid-infrared spectrum (MIR), which is consistent with the combined frequency of the vibration of hydrogen-containing groups (O-H, N-H, and C-H) in organic molecules and the absorption region of each order of frequency doubling. By scanning the NIR spectrum of the sample, the characteristic information of the hydrogen-containing groups in the organic molecules in the sample can be obtained [27,28]. The interaction between these substances and light generates specific spectral peaks and bands, which can be used to determine the chemical composition and substance structure. Previous studies have shown that healthy apples have obvious peaks near wavelengths 650, 705, 769, and 800 nm, with the highest transmission peak in the wavelength range of 705–769 nm [29,30]. The transmission peaks of mold heart disease apples are lower than those of healthy apples, and the more severe the disease, the lower the transmission peak. The affected apples gradually spread and decay outward from the ventricles, with the diseased area appearing black or brownish gray, which has a stronger ability to absorb light. The more severe the disease is, the larger the area of heart decay, the stronger the absorption ability, and the less light passes through, resulting in a lower transmission peak [31,32]. This provides a theoretical basis for the application of transmission NIR spectroscopy to distinguish the degree of apple mold heart disease. Table 1 shows applications of NIR spectroscopy.

In order to study the feasibility of using NIR spectroscopy to detect apple mold heart disease, Andrea et al. designed a low-cost NIR spectroscopy device for the early detection of apple mold heart disease. They found that the artificial neural network pattern recognition (ANN-AP) model combined with the infection growth rate showed a predictive accuracy of 100% at round 1 and 97.15% at round 2 for diseased apples. The research results indicate that the model exhibits consistency in infection rate and development on the test set, thus having greater potential for industrial applicability [33]. Tian et al. designed an online NIR transmission spectroscopy system for detecting apple mold heart disease. The spectral data in the wavelength range of 550 to 900 nm was collected, preprocessed, and used to build support vector machines (SVM) local and global models at three different positions of the apple. It was then found that the T2 positions could obtain the most stable transmission spectral information (Figure 2), and the prediction accuracy of the SVM global model in the test set in all three directions is 100%, which is better than that of the SVM local model. Therefore, it was concluded that the global SVM model is a powerful tool for online detection of mold-heart diseased apples by transmission spectroscopy, regardless of NIR spectral detection positions [34]. Li et al. used a Fourier transform NIR spectrometer with a wavelength range of 833–2500 nm combined with principal component analysis (PCA) to conduct diffuse scanning for apples. They used six methods to pretreat the original spectra to reduce the shape similarity of the NIR spectral curves between healthy apples and mold-heart-diseased apples. The Fisher discriminant models were established after vector normalization and PCA pretreatments. The discriminatory accuracy for new apple samples not involved in modeling in the validation set was 87.8%, which indicates that the application of NIR diffuse reflectance spectroscopy technology in the detection of healthy and mold-heart diseased apples is feasible; however, this result was unsatisfactory [35,36]. Lei et al. designed a transmission NIR spectroscopy acquisition system with a wavelength of 200–1100 nm. They combined the Successive Projections Algorithm (SPA) and PCA to create a variety of models to distinguish mold- and heart-diseased apples from healthy apples. Data processing and analysis are shown in Figure 3. It was found that the precisions of the training and test sets of the PCA-SVM model were 99.3% and 96.7%, which were better than those of the SPA-SVM model. These results provided the technical basis for the development of an apple mold heart disease detection device [11]. Zhang et al. designed a diffuse reflection spectroscopy acquisition platform and measured the apple density data. The collected data were analyzed using four models, including partial least squares discriminant analysis (PLS-DA). The results showed that the SVM-density model had the best discriminant accuracy of 95.56% in the test set, which was better than those of the other three models. And this result was significantly better compared with only using density or spectroscopy data collected [14]. The use of a density meter made the detection process more complicated and time-consuming. Nonetheless, this provides new ideas for the development of future detection equipment.

The primary key to NIR analysis is to establish an accurate and anti-interference calibration model. In addition to appropriate chemometrics methods, standardizing sample spectral collection methods and selecting representative calibration sample sets are also key factors in NIR quantitative analysis. The laboratory-level NIR spectrometer has high detection accuracy and good stability, but its large size, inconvenient portability, and high price limit its application and promotion in mold and heart disease detection.

#### 2.1.2. HSI Technology

Compared with NIR spectroscopy technology, HSI technology has unique advantages. HSI technology is also a non-destructive testing technology that integrates image information and spectral information; it can simultaneously obtain information from multiple optical spectral bands. HSI can obtain more complex fingerprint features, and its image information can reflect external features such as early mold heart disease that are difficult to detect by machine vision. Spectral information can reflect the internal component information of the tested sample and can reflect the internal and external quality information of apples from multiple dimensions. Briefly, the internal chemical compositions and structural differences of apple samples are reflected by spectral information, while the external features such as sample shape, damage, surface defects, and color are reflected by image information [13].

Apple mold heart disease affects the internal organizational structure, thereby affecting apple hardness changes. Based on this principle, HSI technology has the ability to detect mold and heart disease. The moldy parts inside the apple are irregular, and there may be deviations from the center of the moldy parts. Surface defects and other damages on the apple can also affect the detection result. When the apple is placed vertically, the calyx and stem may also affect the transmitted spectral information [28]. Table 2 shows applications of HSI technology. Liu et al. found that when the incidence of mold heart disease is greater than 10% and less than 14%, only the fused spectral morphology feature model can distinguish 100%, and the accuracy of the fruit diameter correction model and detection direction compensation model is higher than that of feature band modeling. When the severity of the disease is less than 10% but greater than 6%, only the fusion spectral morphology feature model has significantly higher discrimination accuracy than other models and can still reach 95.7%. When the severity of the disease is less than 6%, the discrimination accuracy of the detection direction compensation model is 75.0%, the discrimination accuracy of the fruit diameter correction model is 83.3%, and the discrimination accuracy of the fusion spectral morphology feature model can still reach 95.8%. The results indicate that when the degree of disease is less than 10%, the model fused with spectral morphology features significantly improves the accuracy of discrimination, indicating that the model fused with spectral morphology features can effectively express the spectral differences between mold heart disease apples and healthy apples, thereby improving the ability of the model to recognize mild mold heart disease apples. Li et al. used HSI technology in the 874–1734 nm region to detect cherry maturity, and principal component regression (PCR) and partial least squares regression (PLSR) were used to build prediction models of the contents of soluble solid content (SSC) and pH value. Classification of fruit maturity stages was analyzed using linear discrimination, obtaining a classification accuracy of 96.4% [37]. This indicates that it is feasible to detect the quality of fruits using HSI technology. Wang et al. built a HSI non-destructive detection system for apple internal mechanical damage. The convolutional neural network (CNN) model was established, combined with deep learning (DL) technology and a K-cross validation model. The fine-tuned ResNet/ResNeXt obtained average classification accuracy of 0.8844 and 0.8952, respectively. Two DL models obtained better classification accuracy than the traditional machine learning models. ResNet and ResNeXt classifications for each testing sample only took 5.2 ms and 6.5 ms [38], indicating the potential for online fruit sorting of HSI in detecting internal mechanical damage and apple mold heart disease.

The difficulty in the application of HSI technology lies in its high spectral dimension and severe data redundancy, resulting in high computational complexity. Moreover, image information is susceptible to external noise interference, resulting in spectral mixing between its background and abnormal spectra, then leading to content distortion. Therefore, reducing spectral dimensions, streamlining data content, and weakening interference factors have become the development directions of HSI technology. At present, there is no research on apple mold heart disease detection using HSI technology. Many researchers use spectral technology to identify apple mold heart disease but only rely on spectral information, which affects the predictive accuracy. Therefore, providing a detection method or system for apple mold heart disease based on HSI is an urgent problem for technicians in this field.

#### 2.1.3. Raman Spectroscopy Technology

Raman spectroscopy is a vibration spectrum technology based on Raman scattering, which analyzes the scattering spectra with different frequencies from the incident light and enhances the Raman signal through the nanostructure of precious metals to obtain molecular vibration, rotation, and other information [39,40]. The principle of Raman spectrum detection technology is shown in Figure 4. Through molecular vibration and rotation, structure and other relevant information are obtained for detection [27]. The principle of using Raman spectroscopy to detect fruit and vegetable quality is to determine the organic content, chemical composition, or structure of the substance combined with different models [41,42].

Raman spectroscopy has no reports of the detection of apple mold heart disease. However, Raman spectroscopy has already shown great potential for the quality detection of fruits and vegetables. Table 3 shows applications of Raman spectroscopy technology. Josu et al. used Raman technology and confocal Raman microscopy with two excitation laser wavelengths (514 nm and 785 nm) to extract the main information about organic composition in fruit samples. The study shows that confocal Raman microscopy using a 514 nm excitation laser wavelength is a good choice for observing Raman bands and other vibration modes associated with fruit skin compounds [43]. Fan et al. applied a portable surface enhancement Raman scattering (SERS) spectrometer with a 200~2000 cm^−1^ wavelength, a 780 nm laser source, and 100 mW power to perform the measurement of the apple pesticide content. The experimental data were pre-processed by PCA and then developed using PLSR and support vector regression (SVR) algorithms, respectively. The results showed that the SVR model had the best performance, and the correlation coefficient (*R*_p_) in the prediction set was as high as 0.986. This indicates that Raman spectroscopy is a fast and reliable, non-destructive detection technology [44]. Josu et al. built a portable Raman spectroscopic system that consisted of a Raman microprobe, a microcamera, a near-infrared laser with a wavelength of 785 nm, and a power of 350 mW. This system was used to measure the internal maturity of samples within the spectrum range of 100~3000 cm^−1^. The band intensity of the spectral data of the maturity in each maturity period was obtained and analyzed, and they found that the representative spectra were basically consistent with the data measured by the confocal Raman microscope in the laboratory. Apple mold heart disease also represented the internal quality; this proved the feasibility of using Raman spectroscopy as an objective, real, and efficient tool to detect apple mold heart disease [45]. Liu et al. obtained Raman spectra of citrus leaves and identified them as mild, moderate, severe, nutrient deficient, and normal through polymerase chain reaction (PCR). The PLS-DA model fitted with a quadratic polynomial had the best prediction performance, with a predicted *R*_p_ of 0.98 [46]. Guo et al. selected five putrefactive fungi that are prone to apple disease and developed a surface-enhanced Raman spectroscopy based on the gold nanorod substrate method to collect Raman fingerprints of apple fungal spores. The established PCA-LDA model achieved a predictive accuracy of 98.31% [47].

Raman spectroscopy can complement NIR spectroscopy and detect information that cannot be captured by NIR spectroscopy. Raman spectroscopy has a larger detection range and higher detection efficiency compared with NIR spectroscopy, but there are also shortcomings: (1) Compounds with Raman activity are prone to emitting fluorescence, causing Raman signals to be covered and interfering with Raman spectroscopy analysis; (2) Entering any substance in the tested system will contaminate the system and cause significant errors. Therefore, enhancing the Raman signal strength and improving the fault tolerance of the Raman spectroscopy system can be used to improve the detection accuracy and stability of apple mold heart disease [48].

### 2.2. Electronic Nose Technology

Electronic nose detection technology is an intelligent bionic olfactory technology with the advantages of low operation difficulty and high detection sensitivity [49]. The electronic nose detection process is shown in Figure 5. It consists of a gas sensor array, a signal processing unit, and a pattern recognition unit, which can collect gas signals for gas composition analysis and form a feature map [50,51]. The flavor compounds in apples are a mixture of different volatile components such as alcohols, aldehydes, ketones, and so on, which can objectively reflect the quality characteristics of apples. By analyzing their response spectra, it is possible to identify apple diseases, freshness, decay, and damage.

In recent years, electronic noses have been mainly applied in food and other fields, and electronic nose technology is relatively mature, which makes up for the shortcomings of manual evaluation and physical and chemical analysis and provides great convenience for detecting the quality of food and agricultural products [52,53]. Compared with the information stored in the database, it is easy to operate and carry. The combination of an electronic nose, gas chromatography, and mass spectrometry (GC-MS) can evaluate the aroma changes of apples during storage. Apples release 75 volatile components during low-temperature storage, mainly hydrocarbons, aldehydes, ketones, esters, alcohols, acids, terpenoids, and other substances. During storage, the relative content of esters and alcohols released from the apples was relatively high. The content of volatile substances produced by apples shows a significant upward or downward trend during 90 days of storage and mostly shows a significant downward trend at the end of storage. Both produce more ester and alcohol-volatile substances during 90 and 120 days of storage, respectively [54,55]. The odor of apples with mold heart disease is different from that of healthy apples, so it can be concluded that it is feasible to establish a non-destructive detection system for apple mold heart disease based on electronic nose technology. Table 4 shows applications of electronic nose technology. Li used electronic nose technology to detect apple quality based on various physical and chemical quality indexes and used a BP neural network to establish prediction models. They obtained a good predictive correlation of *R*^2^ greater than 0.9000 between mold heart disease and apple odor [55]. Gomez et al. applied the electronic nose system to predict the storage time of fruits, combined with linear discriminant analysis (LDA) and PCA. The correlations of 0.733, 0.726, and 0.659 between the measured and predicted values of fruit quality attributes (such as soluble solid content, acidity, and compression force) showed poor to reasonable prediction performance by means of electronic nose signals. This result could be used in further studies to optimize the number of sensors and find better performance [56]. Zhang et al. used different sensors in the electronic nose system to detect whether apples have mold heart disease. The results of PCA, hierarchical cluster analysis (HCA), and orthogonal projections to latent structures discriminant analysis (OPLS-DA) models showed that volatile odor is one of the important criteria for distinguishing diseased apples. The multi-layer perceptron neural network (MLPNN) discriminant model constructed based on the Fisher function and neural network was superior to other models, with a test set of 88.46% [57]. Yang et al. used electronic nose technology to determine apple mold heart disease and constructed three models, including Fisher discrimination, MLP neural network, and RBF neural network models. Among them, the MLP neural network model has the best discrimination effect, with a discrimination rate in the test set of 86.2% [58]. Jia et al. used an electronic nose to collect the flavor information of inoculated and uninoculated Penicillium apples and mixed apples and established prediction models for fresh and moldy apples using four pattern recognition algorithms, including LDA, BPNN, SVM, and Radial Basis Function Neural Network (RBFNN). The results showed that BPNN had the best prediction performance for the testing sets, with prediction accuracies of 90.0% and 72.0% for fresh and moldy apples, respectively [59].

Although electronic nose technology is widely used, there are still many shortcomings in practical applications. For example, the detection speed is not as fast as the spectral technologies, the detection process is easy to be disturbed by environmental noise and other factors, and equipment is easy to age. Therefore, its application ability is limited to a certain extent. The hardware performance, such as the gas sensor, should be improved in the future to improve the detection speed and noise reduction ability.

### 2.3. Acoustic Technology

Acoustic technology uses the acoustic vibration method as the main method to apply external excitation at a fixed speed to the study object. Due to the differences in the study object, the generated acoustic and vibration response signals will also be different. After collecting signals by an audio collector, the relationship between fruit quality and acoustic and vibration signals can be obtained by means of the control system [60,61].

Table 5 shows applications of acoustic technology. Diezma-Iglesias et al. developed a simple acoustic and vibration detection system to detect the maturity of fruits. The maturity detection of fruits was correlated with the acoustic and vibration frequencies (between 85 and 160 Hz), and the used band magnitude (BM) showed a good prediction effect with a classification accuracy of 89.07%. The result indicates that the acoustic technology is feasible for the maturity detection of fruits [62]. Zhao et al. developed an acoustic non-destructive testing system by using a symmetric polar coordinate analysis method combined with a CNN-SVM model. The optimal parameters of the symmetrized dot pattern (SDP) transformation of acoustic and vibration signals were l = 25 and ξ = 50°. Figure 6 shows the construction flowchart of the early detection model of apple mold heart disease. It was concluded that the ResNet50-SVM-gaus model had the highest discrimination rate; the discrimination accuracy in the testing set of early mold heart diseased apples was 88.89%, and the overall discrimination accuracy in the testing set was up to 96.97% for early mold heart diseased apples and healthy apples. This proved that the acoustic vibration method had the potential to detect apple mold heart disease in its early stages [63].

The non-destructive detecting technology based on acoustic characteristics has the advantages of flexibility, convenience, low cost, and strong adaptability, which has great development space [64]. The disadvantage of an acoustic detection system is that it is not easy to install and debug and is easily disturbed by environmental noise, which affects the free vibration of fruit. Therefore, the relevant system is mostly static and cannot be carried out. In future studies, a noise reduction module can be developed to enable dynamic monitoring, thereby enhancing adaptability [65].

### 2.4. Dielectric Properties Technology

Dielectric properties, including capacitance, resistance, inductance, and so on, vary depending on their components and organizational states. The fruit dielectric electrons are strongly bound by the nucleus, resulting in the formation of internal electric fields. Dielectric technology usually relies on two electrode plates to clamp fruits to obtain the dielectric characteristics in the conductive state. The microscopic changes can be observed between the fruit’s internal quality and electronic microscopic changes. Dielectric characteristic non-destructive detection technology is based on this property [32,66]. The detection based on dielectric properties is a technology that utilizes the dielectric properties of fruits to detect fruit sugar content, SSC, moisture content, disease, hardness, pH value, and other aspects for non-destructive testing, which has the advantages of fast (short measurement and preparation time), convenience (low requirements for the tested substance), and non-destructive (non-destructive online monitoring) [67,68,69,70,71,72,73].

Moisture content, temperature, internal density, and maturity of fruits and vegetables can directly affect the internal capacitance and resistance values, thus affecting their dielectric characteristics. Table 6 shows applications of dielectric properties technology. Khaled et al. analyzed several fruit and vegetable qualities based on dielectric sensing, and the results showed that dielectric properties were strongly correlated with qualities [74]. Li et al. developed an apple dielectric parameter acquisition system with nine kinds of frequencies based on dielectric characteristics. The same index would produce large data differences due to the different frequencies, so the collected data were flattened and preprocessed to ensure the elimination of influences from different dimensions. The modeling results showed that the random forest (RF) neural network model has the best discrimination effect with an accuracy of 95.17% in the testing set. This indicates that dielectric technology combined with an RF model can effectively detect apple mold heart disease [75]. Wang et al. combined PCA with Fisher discriminant, radial basis function neural network (RB-FNN), and multiple perceptron neural network (MPNN) models to detect apple mold heart disease. The ex9periment found that the correct recognition rate of apple mold heart disease reached 100% using a low-frequency loss factor combined with an artificial neural network (ANN), indicating that this technology can be used for the detection of apple mold heart disease [76]. As the degree of mold heart disease increases, the degree of apple decay intensifies. Xu et al. studied the relationship between apple quality and electrical characteristic parameters and found that the relative dielectric constant and equivalent capacitance of rotten apples significantly increased. When the apple diameter is between 7 and 8.5 cm, the diameter of the rotten part is greater than 2 cm, and the measurement frequency of the capacitance sensor is 1 kHz, the classification accuracy can reach over 80%, which indicates dielectric properties technology can effectively detect rotten apples [77]. Euring et al. determined whether apple decay and the degree of decay were related by measuring its voltage and impedance; their correlation of 0.9048 indicated that the voltage increases as the degree of apple decay increases. Dielectric property technology can distinguish between healthy and rotten apples and reflect the degree of apple decay [78].

However, the current research mainly focuses on the influences of the frequency of detection signals, the temperature, and the moisture content on the dielectric characteristics. Future research can optimize dynamic detection, the structure, and the parameters of the dielectric feature detection system to improve sensitivity and reduce dependence on the electrode plate.

### 2.5. Magnetic Technology

Nuclear magnetic resonance (NMR) detection technology is mostly low-field nuclear magnetic resonance (LF-NMR) technology, which means that different radio frequency pulses are applied to the sample in a constant magnetic field so that the hydrogen proton resonates, attenuates, and converges to present different signals. These signals are processed to obtain different graphs or images, and the samples are analyzed through changes. The system reflects the molecular dynamic information of the tested sample [79,80,81].

NMR technology has been applied to the food and agriculture industries in recent years. Table 7 shows applications of NMR technology. With the development of technology, miniaturization and simplification of equipment can become the future development direction. In the future, NMR technology will have great economic value and commercial potential. Lammertyn et al. used magnetic resonance to detect mold and heart disease in fruits. Two kinds of tomography techniques—magnetic resonance imaging (MRI) and X-ray computer tomography (X-ray CT)—were used to monitor fruits stored under mutagenic conditions for a period of 6 months, and this study proved that mold heart disease can be detected by both methods. However, the affected and unaffected tissues in MR images have high contrast, and the gas exchange characteristics are related to the environment [82]. Suchanek et al. used low-field NMR technology combined with multi-layer spin echo sequence (MSE) to obtain h-MR images to detect internal tissue changes in fruit, used ImageJ software (ImageJ 1.45, 2011, National Institutes of Health, Germany) for image analysis and data processing, and distinguished brown tissue from normal tissue with transverse relaxation time (T2). The results showed that there is a small difference between the calculated values of the two groups of fruit images, which confirms the high accuracy of the applied method [83]. Zhou et al. used an MRI scanner to obtain coronal images of the pear. The image was preprocessed using Otsu threshold segmentation, binarization, and boundary extraction, and corner detection was used to determine minor damage. The results showed that the testing accuracy of MRI in detecting surface minor damage to pear was 92.1%, and normal and deformed fruits could be 100% recognized [84]. Chen et al. studied the changes and migration patterns of internal moisture in cherries. Cherries stored at room temperature for 24, 120, 216, 312, 408, and 480 h were detected using LF-NMR technology, and the longitudinal cross-section of the cherry center was obtained. From the images, it can be clearly seen that water content decreases with increasing storage time, and the contours of cherry peel, stone, and flesh gradually become apparent with increasing storage time [85]. Due to the influence of sugar content on the free induction decay signal (FID), lateral relaxation time (T2), and longitudinal relaxation time (T1), Xiong et al. successfully used LF-NMR technology to determine the sugar content in different parts of apples with a correlation coefficient of greater than 0.99, a maximum relative standard deviation of 4%, and a minimum of 1%, indicating high testing accuracy. However, the premise of this method is to shorten the relaxation time of water by adding manganese chloride relaxants, and for the entire apple, adding relaxants is not feasible. Therefore, LF-NMR technology is temporarily unable to complete the sugar content testing of the entire apple [86,87].

NMR technology is a convenient and fast detection technology that does not damage the appearance of the tested apple, nor does it pose potential radiation hazards, and its strong penetration is not limited by the thickness of the apple skin. However, when using NMR detection for different fruits, corresponding NMR studies need to be conducted, and the high cost of NMR instruments limits the application of NMR to some extent [88].

## 3. Research Development Trends and Prospects

With the development of science and technology, the above technologies will become the main force in the field of rapid, non-destructive, intelligent detection of apple mold heart disease in the future. The researchers can choose different technologies according to specific application scenarios. Rapid, non-destructive detection technology brings convenience to researchers, but at the same time, its disadvantages cannot be ignored.

Firstly, due to the heavy equipment and the need to build a variety of models, the development of NIR spectroscopy technology has fallen into a bottleneck. The optimization of equipment and the establishment of complicated models can make greater contributions to this field in the future. Secondly, there are few studies on HSI technology and Raman spectroscopy technology in apple mold heart disease, and their potential has not been fully explored. Thirdly, the electronic nose is greatly disturbed by unexpected odors and other factors, leading to a limited detection ability. Detection ability can be improved through the improvement of software and hardware. Fourthly, the adaptability of acoustic technology is poor due to environmental noise, and it cannot achieve dynamic detection, so the development of portable hardware and the improvement of noise reduction systems have become more important. Fifthly, the difficulty of dielectric technology lies in being easily influenced by apple attributes, making it difficult to obtain quality data with significant differences. Finally, magnetic technology is still being applied in the laboratory and has not yet entered the apple mold heart disease detection market.

The high efficiency of data processing, miniaturization, and simplification of equipment are the future development directions. Due to the rapid development of artificial intelligence (AI), computing power and intelligent algorithms are being enhanced and supplemented as never before. At present, the fusion of AI and rapid non-destructive detection technology is limited to several conventional modeling algorithms. With the development of network technology, AI algorithms can be combined with Big Data technology to integrate multi-source information for processing large amounts of data and then detecting apple mold heart disease. With the rapid development of technology and the support of national policies, it is believed that in the near future, more mature rapid non-destructive detection technology is expected to become a powerful tool in the application of apple mold heart disease detection.

## Figures and Tables

**Figure 1 molecules-28-07966-f001:**
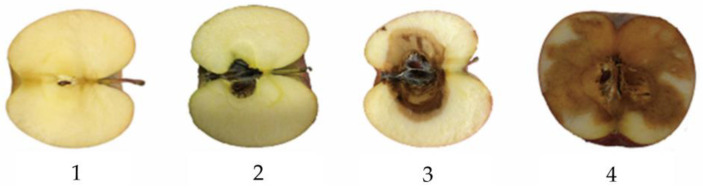
Sample map of different severity of disease. This figure shows the severity of different diseases. Note: 1. Healthy apple; 2. Mild mold heart apple; 3. Moderate mold heart apple; 4. Severe mold on the heart apple.

**Figure 2 molecules-28-07966-f002:**
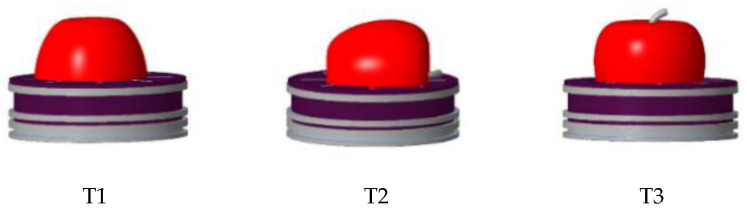
Detection direction of T1, T2, and T3 apples. This image shows the different detection directions for apples.

**Figure 3 molecules-28-07966-f003:**
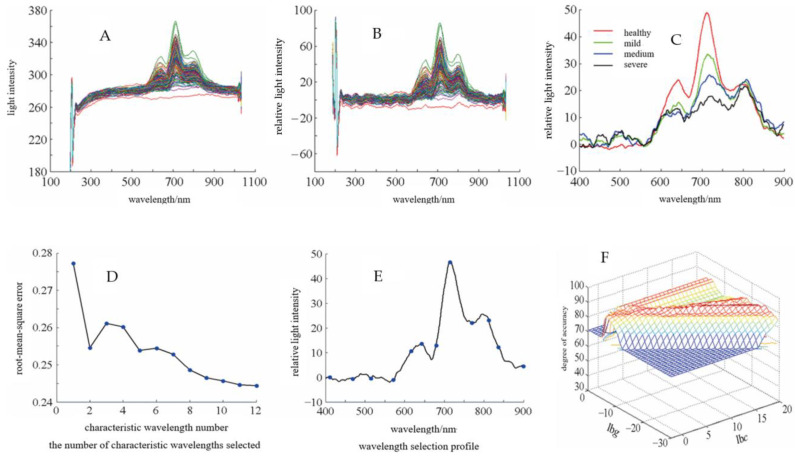
Data processing and analysis. (**A**) Original projected energy. (**B**) After deducting the dark light spectrum (**C**) After MSC preprocessing. (**D**) Number of selected characteristic wavelengths. (**E**) Wavelength selection distribution map. (**F**) Optimization of parameters c and g for SVM.

**Figure 4 molecules-28-07966-f004:**
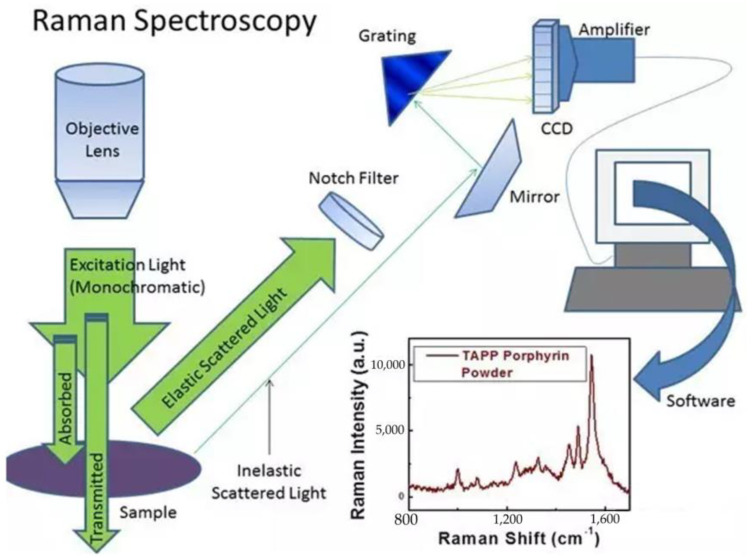
Principles of Raman spectrum detection technology. This is the schematic diagram of Raman spectroscopy detection technology.

**Figure 5 molecules-28-07966-f005:**
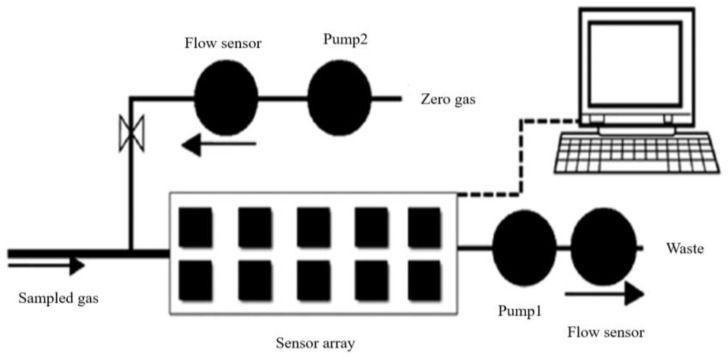
The electronic nose detection process. This image shows the process of electronic nose detection.

**Figure 6 molecules-28-07966-f006:**
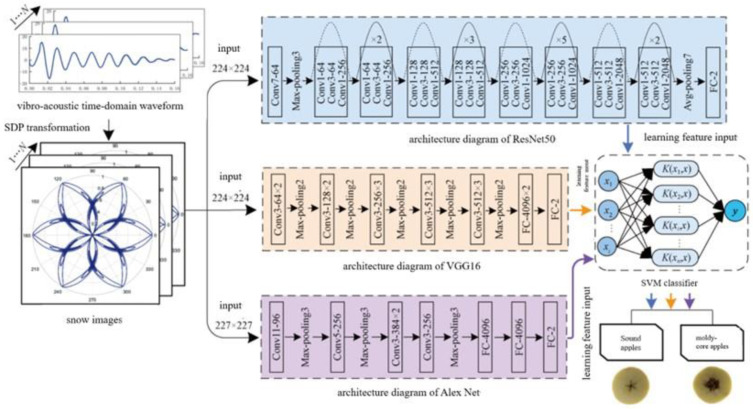
Flow chart of the model construction for the early detection of moldy apple core. This figure shows the process of constructing an early detection model for apple moldy kernels.

**Table 1 molecules-28-07966-t001:** Research on diseases based on NIR spectroscopy technology.

Technology	Object and Indicators	Data Processing	Validation Methods and Parameters
NIR	Apple mold heart disease	S-G, MSC, PCA, SVM	The accuracy of the test set of SVM was 96.7% [11]
	Apple mold heart disease	MSC, SNV, PCA, SVM	The test accuracy of SVM was 90.20% [12]
	Apple mold heart disease	SNV, PLS-DA, SVM	The accuracy of the SVM density model was 95.56% [14]
	Apple mold heart disease	ANN-AP	The prediction accuracy of ANN-AP was 97.15% [33]
	Apple mold heart disease	SGS, normalization, SVM	The prediction accuracy of the SVM global model in test sets in all three directions was 100% [34]
	Apple mold heart disease	Vector normalization, PCA, Fisher	The verification accuracy of Fisher was 87.8% [35,36]

**Table 2 molecules-28-07966-t002:** Research on diseases and quality based on HSI technology.

Technology	Object and Indicators	Data Processing	Validation Methods and Parameters
Spectral Shape Features	Apple mold heart disease	SNV, MSC, NSID, SVM	When disease incidence was greater than 10% and less than 14%, the NSID-SVM accuracy was 100% [28]
HSI	Cherry maturity	PCR, PLS, LDA	The classification accuracy in the test set of the LDA model was 96.4% [37]
HSI	Apple internal mechanical damage	CNN, DL, K-cross validation	Classification accuracy of ResNet/ResNeXt were 0.8844, 0.8952 [38]

**Table 3 molecules-28-07966-t003:** Research on quality is based on Raman spectroscopy.

Technology	Object and Indicators	Data Processing	Validation Methods and Parameters
Raman spectroscopy	Apple pesticide content	PCA, PLSR, SVR	The *R*_p_ in the test set of SVR was 0.986 [44]
	Citrus leaf disease	PCR, PLS-DA	The *R*_p_ of PLS-DA was 0.98 [46]
	5 types of apple spoilage fungi	PCA, LDA	The accuracy in the test set of PCA-LDA was 98.31% [47]

**Table 4 molecules-28-07966-t004:** Research on diseases and quality based on the electronic nose.

Technology	Objects and Indicators	Data Processing	Validation Methods and Parameters
Electronic nose	Apple mold heart disease	BPNN	The *R*^2^ of BPNN was greater than 0.9000 [55]
	Storage time for fruits	LDA, PCA	The correlations of PCA were 0.733, 0.726, and 0.659 [56]
	Apple mold heart disease	PCA, HCA, OPLS-DA, MLPNN	The accuracy in the test set of MLPNN was 88.46% [57]
	Apple mold heart disease	Fisher, MLP	The recognition rate of the MLP test set was 86.2% [58]
	Moldy apples	LDA, BPNN, SVM, RBFNN	The test set accuracy of BPNN was 72.0% [59]

**Table 5 molecules-28-07966-t005:** Research on diseases and quality based on acoustic technology.

Technology	Object and Indicators	Data Processing	Validation Methods and Parameters
Acoustic	Fruit maturity	Vibration frequency (85–160 Hz)	The test set accuracy was 89.07% [62]
	Apple mold heart disease	SDP-CNN-SVM	The overall discrimination accuracy in the testing set of ResNet50-SVM-gaus was 96.97% [63]

**Table 6 molecules-28-07966-t006:** Research on diseases based on Dielectric Properties.

Technology	Objects and Indicators	Data Processing	Validation Methods and Parameters
Dielectric Properties	Apple mold heart disease	PCA, RF	The accuracy in the test set of RF was 95.17% [75]
	Apple mold heart disease	PCA, RB-FNN, MPNN, ANN	The accuracy in the test set of ANN was 100% [76]
	The degree of apple decay	none	The correlation was 0.9048 [78]

**Table 7 molecules-28-07966-t007:** Research on diseases based on Magnetic technology.

Technology	Objects and Indicators	Data Processing	Validation Methods and Parameters
Magnetic	Fruit mold heart disease	MRI, X-ray	Affected and unaffected tissues in MR images have high contrast [82]
	Internal tissue changes in fruit	MSE, ImageJ 1.45 software	There is a small difference between the calculated values of the two groups of fruit images [83]
	Pears damage	Otsu threshold segmentation, binarization, boundary extraction	The accuracy in detecting surface minor damage was 92.1% [84]
	Apples sugar	FID, lateral relaxation time, and longitudinal relaxation time	The correlation coefficient was greater than 0.99 [86,87]

## Data Availability

The data presented in this study are available on request from the corresponding author.

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
