# Peer review of "Research Progress of Rapid Non-Destructive Detection Technology in the Field of Apple Mold Heart Disease"

_molecules, 2023, doi:10.3390/molecules28247966_

Round 1

Reviewer 1 Report

Comments and Suggestions for Authors

In this article, a review of non-destructive detection technology for apple mold heart disease is presented. The topic is quite interesting; however, the collected data that are discussed in the review is shallow and does not have enough details. More details about apple mold heart disease, its effect, and detectable changes by the analytical techniques should be added. The article needs to be revised in the light of the following comments.

1- A deep explanation of the chemical and physiological changes that occur in apple mold heart disease should be explained in the introduction part.

2- The reason for the dangerous effects of apple mold heart disease, including cancer, teratogenicity, and infertility, should be explained.

3- The authors need to discuss what NIR detects exactly in the apple mold heart disease. Is it special chemical compounds that increase in that disease that change the NIR spectra? What are these chemicals? The same also should be discussed in HIS.

4- On page 11 and Table 1, there are many “Error! Reference source not found”. Please correct this.

5- How is the electrical technology diagnostic approach considered non-destructive?

6- Table 1 is neither cited nor discussed in the text!?? The authors need to discuss the difference between the reported methods discussed in Table 1 in detail in Section 3. 

Comments on the Quality of English Language

Moderate editing of the English language is required. 

Author Response

Dear Reviewer:

Thank you very much for giving us an opportunity to make a revision for our manuscript entitled " Research progress of rapid non-destructive detection technology in the field of apple mold heart disease". We have studied comments and revised everything carefully as you suggested. These revisions are highlighted by using coloured (red) text.

Our responses to you are as follows:

  1. In this article, a review of non-destructive detection technology for apple mold heart disease is presented. The topic is quite interesting; however, the collected data that are discussed in the review is shallow and does not have enough details. More details about apple mold heart disease, its effect, and detectable changes by the analytical techniques should be added. The article needs to be revised in the light of the following comments.

1- A deep explanation of the chemical and physiological changes that occur in apple mold heart disease should be explained in the introduction part.

Response:

Thank you for your suggestion. The relevant contents have been added to the revised manuscript (Lines 35-45, 51-61 in chapter 1 Introduction). The changes have been highlighted with revised format.

2- The reason for the dangerous effects of apple mold heart disease, including cancer, teratogenicity, and infertility, should be explained.

Response:

Thank you for your suggestion. The relevant contents have been added to the revised manuscript (Lines 35-45, 51-61 in chapter 1 Introduction). The changes have been highlighted with revised format.

3- The authors need to discuss what NIR detects exactly in the apple mold heart disease. Is it special chemical compounds that increase in that disease that change the NIR spectra? What are these chemicals? The same also should be discussed in HIS.

Response:

Thank you for your suggestion. NIR spectroscopy is not used to detect a specific chemical substance. Spectral signals reflect the comprehensive information of all chemical substances of apple. Specifically, the detection principle of NIR spectroscopy is based on the vibration of hydrogen containing groups (O-H, N-H, C-H) in organic molecules and the absorption region of each order of frequency doubling. The relevant contents discussed in NIR have been added to the revised manuscript (Lines 113-125 in chapter 2.1.1). The relevant contents discussed in HIS have been added to the revised manuscript (Lines 183-188,194-212 in chapter 2.1.2). The changes have been highlighted with revised format.

  1. 4- On page 11 and Table 1, there are many “Error! Reference source not found”. Please correct this.

Response:

Thank you for your comment. The relevant contents have been revised.

5- How is the electrical technology diagnostic approach considered non-destructive?

Response:

Thank you for your comments. To avoid ambiguity for readers, we have changed “the electrical technology” to “dielectric properties”. Many studies reported that the detection based on dielectric properties is a technology that utilizes the dielectric properties of fruits, such as dielectric constant, inductance, and impedance, to detect fruit sugar content, SSC, moisture content, disease, hardness, pH value and other aspects for non-destructive testing. Dielectric properties have the characteristics of fast (short measurement and preparation time), convenience (low requirements for the tested substance), and non-destructive (non-destructive online monitoring). The relevant contents have been added to the revised manuscript (Lines 387-398 in chapter 2.4). The changes have been highlighted with revised format.

The relevant references are as follows:

SOLTANI M, ALIMARDANI R, OMID M. Evaluating banana ripening status from measuring dielectric properties[J]. Journal of Food Engineering, 2011, 105(4): 625-631. DOI:10.1016/j.jfoodeng.2011.03.032.

VEAL C D, WEBB B K. Use of radio frequency waves for evaluation of tomato maturity[C]// Chicago: The Annual Meeting of ASAE, 1971.

GUO Wenchuan, ZHU Xinhua, NELSON S O, et al. Maturity effects on dielectric properties of apples from 10 to 4 500 MHz[J]. LWT Food Science and Technology, 2011, 44(1): 224-230. DOI:10.1016/ j.lwt.2010.05.032.

NELSON S O, GUO W C, TRABELSI S, et al. Dielectric spectroscopy of watermelons for quality sensing[J]. Measurement Science and Technology, 2007, 18(7): 1887-1892. DOI:10.1088/0957- 0233/18/7/014.

CORTES H, EDGAR V, SUAREZ R, et al. Pitahaya aging diagnostic by impedance/capacitance spectroscopy[J]. Food Analytical Methods, 2015, 8(1): 126-129. DOI:10.1007/s12161-014-9878-7.

GUO Wenchuan, NELSON S O, TRABELSI S, et al. 10-1800-MHz dielectric properties of fresh apples during storage[J]. Journal of Food Engineering, 2007, 83(4): 562-569. DOI:10.1016/ j.jfoodeng.2007.04.009.

6- Table 1 is neither cited nor discussed in the text!?? The authors need to discuss the difference between the reported methods discussed in Table 1 in detail in Section 3.

Response:

Thank you for your comments. In order to facilitate reading and response to the requests of other reviewers, we have created Tables for each technology, and added more contents in each technology section. The changes have been highlighted with revised format.

At last, sincerely thank you for the comments.

Sincerely,

Yanlei Li

Reviewer 2 Report

Comments and Suggestions for Authors

The present review discusses the application of some non-destructive methods for the identification of apple mold heart disease. I think for a good review the cited references must be at least 80 to 100 references.

I think the author must add the discussion of the basic principle the method of identification of apple mold disease (NIR, HIS, Raman etc.), and discuss also the advantages and disadvantages of the methods.

It is well known that the validity and reproducibility of the published methods depends whether the methods are already “completed” validated or not, so it will be nice the authors mention whether the cited publications have been completed validated or not. If yes please describe briefly their methods and the data, and if it is not please also mention it.

I recommend you the make tables for each of the methods used (not as described by table 1).  I think the term “principle method” must be replaced with “data processing”, and “model effect” with “validation methods and parameters”. Please describe complete validation parameters, whether they have done or not. What did you mean by “Error! Reference source not found”?

Author Response

Dear Reviewer:

Thank you very much for giving us an opportunity to make a revision for our manuscript entitled " Research progress of rapid non-destructive detection technology in the field of apple mold heart disease". We have studied comments and revised everything carefully as you suggested. These revisions are highlighted by using coloured (red) text.

Our responses to you are as follows:

The present review discusses the application of some non-destructive methods for the identification of apple mold heart disease. I think for a good review the cited references must be at least 80 to 100 references.

Response:

Thank you for your comment. We have added more references and more discussion contents in the paper. The changes have been highlighted with revised format.

I think the author must add the discussion of the basic principle the method of identification of apple mold disease (NIR, HIS, Raman etc.), and discuss also the advantages and disadvantages of the methods.

Response:

Thank you for your comment. The relevant contents have been added to the revised manuscript (Lines 113-125, 165-170 in chapter 2.1.1\ Lines 183-188, 226-231 in chapter 2.1.2\ Lines 281-289 in chapter 2.1.3\ Lines 299-302, 307-318, 344-349 in chapter 2.2\ Lines 375-381 in chapter 2.3\ Lines 387-398, 426-430 in chapter 2.4\ Lines 474-479 in chapter 2.5). The changes have been highlighted with revised format.

It is well known that the validity and reproducibility of the published methods depends whether the methods are already “completed” validated or not, so it will be nice the authors mention whether the cited publications have been completed validated or not. If yes please describe briefly their methods and the data, and if it is not please also mention it.

Response:

Thank you for your comment. We have carefully read all the cited publications again, confirmed whether the cited publications have been completed validated or not and added validation data in the text and tables. We have provided a summary description of some publications without relevant validation data. The relevant contents have been added to the revised manuscript (Lines 129-132, 138-141, 146-147, 154, 160 in chapter 2.1.1\ Lines 194-212, 215-217, 220-225 in chapter 2.1.2\ Lines 274-280 in chapter 2.1.3\ Lines 323-327, 337-343 in chapter 2.2\ Lines 362-363, 370-372 in chapter 2.3\ Lines 408, 410-425 in chapter 2.4\ Lines 455-473 in chapter 2.5)

I recommend you the make tables for each of the methods used (not as described by table 1).  I think the term “principle method” must be replaced with “data processing”, and “model effect” with “validation methods and parameters”. Please describe complete validation parameters, whether they have done or not. What did you mean by “Error! Reference source not found”?

Response:

Thank you for your comment. We have made new tables for each of the methods used, revised terms and all validation parameters according to your comment. “Error! Reference source not found” is a system error after submitting the paper, and we have modified it.

At last, sincerely thank you for the comments.

Sincerely,

Yanlei Li

Round 2

Reviewer 1 Report

Comments and Suggestions for Authors

The authors have revised the manuscript according to the reviewer's comments appropriately, however, there is still some required minor correction that was not addressed well.

1- The authors' responses to my questions 1 and 2 are not clear enough. Please give examples of the toxins accumulated in the case of apple mold heart disease. 

2- There is still "Error! Reference source not found" in Tables 1 and 5. Please check and correct this error.

Author Response

Dear Reviewer:

Thank you very much for giving us an opportunity to make a minor revision for our manuscript entitled " Research progress of rapid non-destructive detection technology in the field of apple mold heart disease". We have studied comments and revised everything carefully as you suggested. These revisions are highlighted by using coloured (red) text.

Our responses to you are as follows:

1- The authors' responses to my questions 1 and 2 are not clear enough. Please give examples of the toxins accumulated in the case of apple mold heart disease.

Response:

Thank you for your comment. The relevant contents have been added to the revised manuscript ((Lines 35-44 in chapter 1).

2- There is still "Error! Reference source not found" in Tables 1 and 5. Please check and correct this error.

Response:

Thank you for your suggestion. Due to issues with the publisher's system, an error occurred during upload, and the publisher has provided an explanation and correction.

At last, sincerely thank you for the comments.

Sincerely,

Yanlei Li

Reviewer 2 Report

Comments and Suggestions for Authors

“Error! Reference source not found” were still printed in table 1

Author Response

Dear Reviewer:

Thank you very much for giving us an opportunity to make a minor revision for our manuscript entitled " Research progress of rapid non-destructive detection technology in the field of apple mold heart disease". We have studied comments and revised everything carefully as you suggested.

Our responses to you are as follows:

“Error! Reference source not found” were still printed in table 1

Response:

Thank you for your suggestion. Due to issues with the publisher's system, an error occurred during upload, and the publisher has provided an explanation and correction.

At last, sincerely thank you for the comments.

Sincerely,

Yanlei Li
